# The Influence of Temperature and Community Structure on Light Absorption by Phytoplankton in the North Atlantic

**DOI:** 10.3390/s19194182

**Published:** 2019-09-26

**Authors:** Robert J. W. Brewin, Stefano Ciavatta, Shubha Sathyendranath, Jozef Skákala, Jorn Bruggeman, David Ford, Trevor Platt

**Affiliations:** 1College of Life and Environmental Sciences, University of Exeter, Penryn Campus, Cornwall TR10 9FE, UK; 2Plymouth Marine Laboratory, Plymouth, Devon PL1 3DH, UK; avab@pml.ac.uk (S.C.); ssat@pml.ac.uk (S.S.); jos@pml.ac.uk (J.S.); jbr@pml.ac.uk (J.B.); tplatt@dal.ca (T.P.); 3National Centre for Earth Observation, Plymouth Marine Laboratory, Plymouth, Devon PL1 3DH, UK; 4Met Office, Exeter, Devon EX1 3PB, UK; david.ford@metoffice.gov.uk

**Keywords:** phytoplankton absorption, community structure, temperature, North Atlantic

## Abstract

We present a model that estimates the spectral phytoplankton absorption coefficient ($a_{ph}\left(\lambda \right)$) of four phytoplankton groups (picophytoplankton, nanophytoplankton, dinoflagellates, and diatoms) as a function of the total chlorophyll-a concentration (*C*) and sea surface temperature (SST). Concurrent data on $a_{ph}\left(\lambda \right)$ (at 12 visible wavelengths), *C* and SST, from the surface layer (<20 m depth) of the North Atlantic Ocean, were partitioned into training and independent validation data, the validation data being matched with satellite ocean-colour observations. Model parameters (the chlorophyll-specific phytoplankton absorption coefficients of the four groups) were tuned using the training data and found to compare favourably (in magnitude and shape) with results of earlier studies. Using the independent validation data, the new model was found to retrieve total $a_{ph}\left(\lambda \right)$ with a similar performance to two earlier models, using either in situ or satellite data as input. Although more complex, the new model has the advantage of being able to determine $a_{ph}\left(\lambda \right)$ for four phytoplankton groups and of incorporating the influence of SST on the composition of the four groups. We integrate the new four-population absorption model into a simple model of ocean colour, to illustrate the influence of changes in SST on phytoplankton community structure, and consequently, the blue-to-green ratio of remote-sensing reflectance. We also present a method of propagating error through the model and illustrate the technique by mapping errors in group-specific $a_{ph}\left(\lambda \right)$ using a satellite image. We envisage the model will be useful for ecosystem model validation and assimilation exercises and for investigating the influence of temperature change on ocean colour.

## 1. Introduction

Light absorbed by phytoplankton is either converted to heat or used in photosynthesis. The conversion to heat helps modulate the physical structure of the ocean’s surface layer [1,2]. Photosynthesis converts inorganic to organic carbon, modifying the CO${}_{2}$ concentration and pH of the water, and providing energy to roughly half of the life on our planet. Consequently, light absorption by phytoplankton plays an important role in the functioning of the ocean.

The phytoplankton absorption coefficient ($a_{ph}\left(\lambda \right)$, where λ is the wavelength of light), is used to represent the amount of light absorbed by phytoplankton per unit length. It is a fundamental quantity in physical and biogeochemical ocean models, altering underwater spectral light transmission [3,4,5,6] and the photosynthetic response of phytoplankton to available light [7,8,9]. The magnitude and spectral shape of $a_{ph}\left(\lambda \right)$ are controlled by phytoplankton biomass, size and type. Consequently, routine measurements of $a_{ph}\left(\lambda \right)$ are used to monitor the amount, composition and size structure of phytoplankton in aquatic systems [10,11,12,13,14,15,16].

Historically and conventionally, $a_{ph}\left(\lambda \right)$ has been modelled as a function of the total chlorophyll-a concentration (*C*, representing the sum of mono- and divinyl-chlorophyll-a, chlorophyllide-a, and the allomeric and epimeric forms of chlorophyll-a), the main photosynthetic pigment in all phytoplankton and a measure of phytoplankton biomass. Nonlinearity in this relationship, which varies with wavelength, is indicative of changes in phytoplankton size and pigment composition [17,18,19,20,21,22]. Empirical models that derive $a_{ph}\left(\lambda \right)$ from *C* have been proposed, including: power-law functions [22,23,24,25,26,27]; hyperbolic tangent functions [28]; polynomial functions [29]; and Michaelis-Menten-type functions [6,30].

Theoretical approaches have also been proposed, for example, those that express $a_{ph}\left(\lambda \right)$ as the contribution of different populations of phytoplankton in the water [10,14,31,32,33], such that $a_{ph}\left(\lambda \right)$ can take the form
(1)$$\begin{array}{c} \begin{array}{c} a_{ph}\left(\lambda \right)=\sum_{i=1}^{N}a_{i}^{*}\left(\lambda \right)C_{i}, \end{array} \end{array}$$
where *i* is the population, *N* is the number of different populations, $a_{i}^{*}\left(\lambda \right)$ is the chlorophyll-specific absorption coefficient for population *i*, and $C_{i}$ is the chlorophyll-a concentration for population *i* (Table A1 in the Appendix A to this manuscript defines all symbols used). Unlike empirical models [30], the parameters of Equation (1) have clear interpretation and the approach ensures plausible values of $a^{*}\left(\lambda \right)$ at extreme chlorophyll-a concentrations, since the range of values of $a^{*}\left(\lambda \right)$ is bounded by the values associated with the populations. Models with two [32] and three [14,31,33] populations have been proposed, typically partitioning populations of phytoplankton according to size ranges, since $a^{*}\left(\lambda \right)$ is known to change with cell size [10].

Approaches have also been proposed that tie changes in $C_{i}$ with *C* [32,34], meaning that $a_{ph}\left(\lambda \right)$ can be derived from Equation (1) as a function of *C*, by relating $C_{i}$ to *C*, if a priori information exists on $a_{i}^{*}\left(\lambda \right)$ [14,31,32]. It is widely recognised that temperature is a useful variable for predicting phytoplankton community structure [35,36,37,38,39,40,41,42,43,44] and consequently, $a^{*}\left(\lambda \right)$ [45]. Recently, Brewin et al. [46] proposed a model that estimates $C_{i}$ for four populations of phytoplankton (picophytoplankton, nanophytoplankton, dinoflagellates, and diatoms) as a function of *C* and sea surface temperature (SST) in the North Atlantic, both of which can be retrieved through satellite remote sensing. The approach has proven useful for satellite data assimilation into multi-phytoplankton ecosystem models [47,48], although assimilating the optical properties of the phytoplankton groups directly (i.e., $a_{ph,i}\left(\lambda \right)$ rather than $C_{i}$) could be more beneficial [49].

In this paper, using the approach of Brewin et al. [46], we extend the three-population absorption model of Brewin et al. [31] to four populations of phytoplankton. Our purposes for developing this new model were twofold: (1) to estimate $a_{ph}\left(\lambda \right)$ for four phytoplankton groups (dinoflagellates, diatoms, nanophytoplankton, and picophytoplankton) that match those simulated in a state of the art marine ecosystem model (the European Regional Seas Ecosystem Model, ERSEM), using remotely-sensed input (*C* and SST), and quantify uncertainties in these estimates for use in a future optical data-assimilation experiment using ERSEM; and (2) investigate how temperature change may influence ocean colour in the context of climate change.

The new model yields total and population-specific $a_{ph}\left(\lambda \right)$ as a continuous function of *C* and SST. Data from the North Atlantic containing $a_{ph}\left(\lambda \right)$, *C* and SST measurements are separated into parameterisation and validation data. The parameterisation data are used to yield $a_{i}^{*}\left(\lambda \right)$ values for each of the four populations, which are subsequently compared with results from earlier studies. The performance of the model, when used to retrieve $a_{ph}\left(\lambda \right)$ for a given total chlorophyll concentration, is compared with those of two other models (a power-law model and a three-population model) using the independent validation data. An error-propagation method is proposed to quantify uncertainty in population-specific $a_{ph}\left(\lambda \right)$ derived from the four-population absorption model, and we integrate the four-population absorption model into a simple (case-1) model of ocean colour, to illustrate expected changes in the blue-to-green ratio of remote-sensing reflectance with changes in *C* and SST.

## 2. Methodology

### 2.1. Study Area

Our study area was the North Atlantic Ocean, spanning 100 ${}^{\circ }$ W to 13 ${}^{\circ }$ E and 20 ${}^{\circ }$ N to 66 ${}^{\circ }$ N (Figure 1). This region encompasses the Copernicus Marine Environment Monitoring Service (CMEMS) Ocean Colour Thematic Assembley Centre (OCTAC) Atlantic (ATL) region, the CMEMS Marine Forecasting Centre (MFC) of the North West Shelf-Seas (NWS) and Ireland-Biscay-Iberia (IBI) regions, as well as the north-west Atlantic and eastern seaboard of North America, covering a wide range of bio-optical environments, from the oligotrophic North Atlantic gyre through to shallower optically-complex shelf seas. The region has been sampled intensively over the past few decades, supports one of the largest spring phytoplankton blooms on the planet [50] and is a region of focus for many marine ecosystem modelling studies (e.g., [51]).

### 2.2. Data

Two bio-optical datasets were utilised in the study: (1) the NASA bio-Optical Marine Algorithm Dataset (NOMAD Version 2.0w APLHA, 18/07/2008 [52,53]); and (2) a dataset compiled from various locations by Shubha Sathyendranath and Trevor Platt at the Bedford Institute of Oceanography [10,32]. Both datasets contained matching measurements of $a_{ph}\left(\lambda \right)$ and total chlorophyll concentration (*C*), the latter derived either from High Performance Liquid Chromatography or using a calibrated Turner fluorometer following extraction of chlorophyll in solvent. For the NOMAD dataset, $a_{ph}\left(\lambda \right)$ was derived by subtracting detrital absorption ($a_{d}\left(\lambda \right)$) from particulate absorption ($a_{p}\left(\lambda \right)$) measurements. Because the two datasets provided measurements of $a_{ph}\left(\lambda \right)$ at slightly different wavebands, 12 common wavebands (412, 443, 490, 510, 520, 550, 560, 620, 665, 670, and 682 nm) were selected for which the two datasets had wavelengths within 1 nm of this common set. The wavebands are also aligned with those of common multispectral ocean colour sensors. Only data within the selected study area were used (see Figure 1) and within the top 20 m of the water column (within the surface mixed-layer depth (rarely <20 m in the open ocean [54]) or within the 1st optical depth as in the case of the NASA NOMAD dataset. For each measurement, SST data were extracted by matching each in situ sample in time (daily temporal match-up) and space (closest latitude and longitude) with daily, 1/4${}^{\circ }$ resolution Optimal Interpolation Sea Surface Temperature (OISST) data (Version 2.0 [55]) acquired from the National Oceanic and Atmospheric Administration (NOAA) website (http://www.esrl.noaa.gov/psd/data/gridded/data.noaa.oisst.v2.highres.html). In total, 1687 measurements of $a_{ph}\left(\lambda \right)$, *C* and SST (median = 11.5 ${}^{\circ }$C, min = 0.0 ${}^{\circ }$C, max = 30.8 ${}^{\circ }$C, 13.6 percentile = 3.4 ${}^{\circ }$C, and 86.4 percentile = 25.1 ${}^{\circ }$C) were available for use, covering all months of the year.

The data were matched to daily, level 3 (4 km sinusoidal projected) satellite chlorophyll data, from version 3.1 of the Ocean Colour Climate Change Initiative (OC-CCI, a merged MERIS, MODIS-Aqua and SeaWiFS product available at http://www.oceancolour.org/), between 1997–2006. Each in situ sample was matched with a single satellite pixel in time (same day) and space (closest pixel with a distance <4 km away). For the 1687 samples, there were 484 corresponding satellite chlorophyll match-ups. These 484 measurements were set aside and used for independent model validation, leaving 1203 measurements for model training (parameterisation). Figure 1 shows the geographic distribution of the training and validation data.

### 2.3. Four-Population Model of Phytoplankton Absorption

Brewin et al. [46] consider the total chlorophyll-a concentration (*C*) as the sum of chlorophyll-a concentrations in picophytoplankton ($C_{1}$), nanophytoplankton ($C_{2}$), dinoflagellates ($C_{3}$) and diatoms ($C_{4}$), such that
(2)$$\begin{array}{c} \begin{array}{c} C=\sum_{i=1}^{4}C_{i}. \end{array} \end{array}$$

The model first utilises two exponential functions [10], where the chlorophyll concentration of picophytoplankton ($C_{1}$, cells <2 μm) and combined pico- and nanophytoplankton ($C_{1,2}$, cells <20 μm) are obtained from
(3)$$\begin{array}{c} C_{1,2}=C_{1,2}^{m}\left[1-\exp\left(-\frac{D_{1,2}}{C_{1,2}^{m}}C\right)\right], \end{array}$$
and
(4)$$\begin{array}{c} C_{1}=C_{1}^{m}\left[1-\exp\left(-\frac{D_{1}}{C_{1}^{m}}C\right)\right]. \end{array}$$

The parameters $D_{1,2}$ and $D_{1}$ determine the fraction of total chlorophyll in the two size classes (<20 μm and <2 μm, respectively) as total chlorophyll tends to zero, and $C_{1,2}^{m}$ and $C_{1}^{m}$ are the asymptotic maximum values for the two size classes (<20 μm and <2 μm, respectively). The chlorophyll concentration of nanophytoplankton ($C_{2}$) and microphytoplankton (combined dinoflagellates and diatoms, $C_{3,4}$) are calculated simply as $C_{2}=C_{1,2}-C_{1}$ and $C_{3,4}=C-C_{1,2}$. Brewin et al. [46] modelled the parameters of Equations (3) and (4) as a function of SST using the following logistic equations,
(5)$$\begin{array}{c} \begin{array}{c} C_{1,2}^{m}=1-\left\{\frac{G_{a}}{1+\exp[-G_{b}\left(\mathrm{SST}-G_{c}\right)]}+G_{d}\right\}, \end{array} \end{array}$$
and
(6)$$\begin{array}{c} \begin{array}{c} C_{1}^{m}=1-\left\{\frac{H_{a}}{1+\exp[-H_{b}\left(\mathrm{SST}-H_{c}\right)]}+H_{d}\right\}, \end{array} \end{array}$$
where $G_{a}$ and $G_{d}$ control the upper and lower bounds of $C_{1,2}^{m}$, $G_{b}$ represents the slope of change in $C_{1,2}^{m}$ with SST, and $G_{c}$ is the SST mid-point of the slope between $C_{1,2}^{m}$ and SST. For $C_{1}^{m}$, $H_{i}$, where i=
*a* to *d*, is analogous to $G_{i}$ for $C_{1,2}^{m}$. The parameters $D_{1,2}$ and $D_{1}$ were expressed as
(7)$$\begin{array}{c} \begin{array}{c} D_{1,2}=\frac{J_{a}}{1+\exp[-J_{b}\left(\mathrm{SST}-J_{c}\right)]}+J_{d}, \end{array} \end{array}$$
and
(8)$$\begin{array}{c} \begin{array}{c} D_{1}=\frac{K_{a}}{1+\exp[-K_{b}\left(\mathrm{SST}-K_{c}\right)]}+K_{d}, \end{array} \end{array}$$
where $J_{a}$ and $J_{d}$ control the upper and lower bounds of $D_{1,2}$, $J_{b}$ represents the slope of change in $D_{1,2}$ with SST, and $J_{c}$ is the SST mid-point of the slope between $D_{1,2}$ and SST. For $D_{1}$, $K_{i}$ is analogous to $J_{i}$ for $D_{1,2}$. Model parameters $G_{i}$, $H_{i}$, $J_{i}$ and $K_{i}$ are provided in Table 1. Finally, microphytoplankton ($C_{3,4}$) is partitioned into dinoflagellates ($C_{3}$) and diatoms ($C_{4}$) according to,
(9)$$\begin{array}{c} \begin{array}{c} \frac{C_{3}}{C_{3,4}}=\frac{1}{1+\exp[-\alpha (\mathrm{SST}-\beta )]}, \end{array} \end{array}$$
where α=0.10 (0.08↔0.13) and β=32.5 (29.7↔36.1). Using Equations (3)–(9), the chlorophyll concentrations for the four groups ($C_{1}$, $C_{2}$, $C_{3}$ and $C_{4}$) can be derived from total chlorophyll (*C*) and SST.

Here, we expand on this approach by modelling $a_{ph}\left(\lambda \right)$ as the contribution of the four different populations of phytoplankton, picophytoplankton ($a_{ph,1}\left(\lambda \right)$), nanophytoplankton ($a_{ph,2}\left(\lambda \right)$), dinoflagellates ($a_{ph,3}\left(\lambda \right)$), and diatoms ($a_{ph,4}\left(\lambda \right)$), according to
(10)$$\begin{array}{c} \begin{array}{c} a_{ph}\left(\lambda \right)=\sum_{i=1}^{4}a_{i}^{*}\left(\lambda \right)C_{i}, \end{array} \end{array}$$
where $a_{i}^{*}$ are the chlorophyll-specific absorption coefficients for each population *i*. To retrieve $a_{i}^{*}$, we used the training (parameterisation) data. By integrating Equations (3)–(9) into Equation (10), and using $a_{ph}$, *C* and SST as inputs, we fitted Equation (10) individually to each wavelength using a non-linear least-squared fitting procedure (Levenberg-Marquardt [56,57], IDL Routine MPFITFUN). We used the method of bootstrapping [58,59] to compute a parameter distribution (1000 bootstraps), and from the resulting parameter distribution median values and robust standard deviations in $a_{i}^{*}$ were obtained, which are provided in Table 2 for each of the 12 wavelengths in the dataset.

### 2.4. Other Models That Relate Phytoplankton Absorption to Total Chlorophyll

Using the independent validation data, the performance of the absorption model developed in the previous section at retrieving total $a_{ph}\left(\lambda \right)$ was compared with two existing phytoplankton absorption models that derive total $a_{ph}\left(\lambda \right)$ as a function of *C*: a power-law model [23]; and a three-population model [31]. The power-law model can be expressed as
(11)$$\begin{array}{c} \begin{array}{c} a_{ph}\left(\lambda \right)=A\left(\lambda \right)C^{\left(1-B(\lambda )\right)}, \end{array} \end{array}$$
where A(λ) and B(λ) are positive, wavelength-dependent parameters. These were taken from Table 2 of Bricaud et al. [23]. The three-population model can be expressed as
(12)$$\begin{array}{ccc} \;a_{ph}\left(\lambda \right) & = & a_{1}^{*}\left(\lambda \right)C_{1}^{m}\left[1-\exp\left(-\frac{D_{1}}{C_{1}^{m}}C\right)\right]+\\ \; & \; & a_{2}^{*}\left(\lambda \right)\left\{C_{1,2}^{m}\left[1-\exp\left(-\frac{D_{1,2}}{C_{1,2}^{m}}C\right)\right]-C_{1}^{m}\left[1-\exp\left(-\frac{D_{1}}{C_{1}^{m}}C\right)\right]\right\}+\\ \; & \; & a_{3,4}^{*}\left(\lambda \right)\left\{C-C_{1,2}^{m}\left[1-\exp\left(-\frac{D_{1,2}}{C_{1,2}^{m}}C\right)\right]\right\}. \end{array}$$

In this approach, dinoflagellates (i=3) and diatoms (i=4) are grouped together as microphytoplankton (i=3,4) and the parameters $D_{1,2}$, $D_{1}$, $C_{1,2}^{m}$ and $C_{1}^{m}$ are fixed and consequently do not vary with SST, in contrast to Equations (5)–(8). Model parameters for Equation (12) were taken from Tables 1 and 2 of Brewin et al. [31] (note in their Table 1 $D_{i}=C_{i}^{m}S_{i}$). The performances of the three models (four-population, three-population and power-law) at estimating $a_{ph}\left(\lambda \right)$ from *C* were evaluated using both in situ and satellite estimates of *C*.

### 2.5. Statistical Tests

Model performance was quantified using the Pearson linear correlation coefficient (*r*) and the root mean square error (Ψ) between the estimated and measured absorption coefficients. The Ψ were computed according to
(13)$$\begin{array}{c} \begin{array}{c} \Psi ={\left[\frac{1}{N}\sum_{i=1}^{N}{\left(X_{i}^{E}-X_{i}^{M}\right)}^{2}\right]}^{1/2}, \end{array} \end{array}$$
where *X* is the variable ($a_{ph}\left(\lambda \right)$) and *N* is the number of samples. The superscript *E* denotes the estimated variable (e.g., from model) and *M* the measured variable (e.g., in situ). All statistical tests were performed in log${}_{10}$ space, considering that $a_{ph}\left(\lambda \right)$ is approximately log-normally distributed [60].

### 2.6. Estimation of Uncertainty in Group-Specific $a_{ph}\left(\lambda \right)$

Because $a_{ph,i}\left(\lambda \right)$ can be expressed as the product of $a_{i}^{*}\left(\lambda \right)$ and $C_{i}$, the relative uncertainty (or relative standard deviation) in $a_{ph,i}\left(\lambda \right)$, denoted as $f_{a,i}\left(\lambda \right)$, can be approximated, based on a Taylor expansion, as
(14)$$\begin{array}{c} \begin{array}{c} f_{a,i}\left(\lambda \right)=\sqrt{f_{a,i}^{*}{\left(\lambda \right)}^{2}+{f_{C,i}}^{2}}, \end{array} \end{array}$$
where $f_{a,i}^{*}\left(\lambda \right)$ and $f_{C,i}$ are the relative uncertainties in $a_{i}^{*}\left(\lambda \right)$ and $C_{i}$, respectively, and *i* is the phytoplankton group. The quantity $f_{a,i}^{*}\left(\lambda \right)$ can be estimated as the standard deviation in $a_{i}^{*}\left(\lambda \right)$ divided by $a_{i}^{*}\left(\lambda \right)$ (Table 2). So with knowledge of $f_{C,i}$ we can compute $f_{a,i}\left(\lambda \right)$ through application of Equation (14).

Here, we illustrate the application of Equation (14) using a satellite image of total chlorophyll (*C*) and SST in the North East Atlantic (8-day chlorophyll (OC-CCI) and SST (NOAA OISST) composite between 17 and 24 June 2008). For each group (picophytoplankton, nanophytoplankton, diatoms, and dinoflagellates), $C_{i}$ and associated log${}_{10}$-transformed per-pixel uncertainties were estimated using the model of Brewin et al. [46]. Briefly, Brewin et al. [46] used an in situ and satellite match-up dataset to compute the log${}_{10}$-transformed Ψ and the bias between in situ and satellite estimates of $C_{i}$, for 14 different optical water types (OWT, see Table 5 of [46]). These values are weighted using satellite estimates of OWT membership [61] to map Ψ and bias in $C_{i}$ for each pixel.

Here, we used the approach of Ciavatta et al. [62] (see their Appendix A) to transform data uncertainties between log and linear space. The $C_{i}$ data were bias-corrected and per-pixel standard deviations were computed in linear-space. We then computed $f_{C,i}$ by dividing the standard deviations by the concentrations. Finally, $f_{a,i}\left(\lambda \right)$ was computed on a per-pixel basis through application of Equation (14) using per-pixel $f_{C,i}$ and estimates of $f_{a,i}^{*}\left(\lambda \right)$ from Table 2 as input.

## 3. Results and Discussion

### 3.1. Model Tuning

To evaluate the tuning of the four-population model, estimates of $a_{ph}\left(\lambda \right)$ from the model, using *C* and SST as input, are plotted against observations of $a_{ph}\left(\lambda \right)$ in the parameterisation data (Figure 2). The model is seen to fit well to the observations, with correlation coefficients ≥0.89, comparable with the results of studies fitting other models to $a_{ph}\left(\lambda \right)$ and *C* data [31,32], and root mean square errors (Ψ) ranging between 0.20 and 0.27 for all wavelengths (Figure 2).

The retrieved $a_{i}^{*}\left(\lambda \right)$ values for each phytoplankton group, computed from fitting Equation (10) to the parameterisation data, are shown in Figure 3a and their spectral forms ($a^{*}\left(\lambda \right)$ normalised at 510 nm) in Figure 3b. Picophytoplankton have the highest $a^{*}\left(\lambda \right)$ and the steepest spectral form. Diatoms display the lowest $a^{*}\left(\lambda \right)$ values and the flattest spectral form. Nanophytoplankton and dinoflagellates $a^{*}\left(\lambda \right)$ lie between picophytoplankton and diatoms. The $a^{*}\left(\lambda \right)$ of all groups have peaks around 443 nm and 670 nm associated with chlorophyll-a absorption. Changes in $a^{*}\left(\lambda \right)$ magnitude and spectral form (shape) from picophytoplankton (smallest cells) to diatoms (large cells) are consistent with changes in size associated with packaging and pigment composition [17,18,19,20,21,22,63,64].

Retrieved picophytoplankton $a^{*}\left(\lambda \right)$ compare well with results from other studies (Figure 3c). Retrieved values are higher than those of Ciotti and Bricaud [65], but in good agreement with Brewin et al. [31], Devred et al. [14], and Uitz et al. [33], with confidence intervals overlapping for most wavelengths. Agreement with the Brewin et al. [31] and Devred et al. [14] studies is not surprising, considering data used here are not entirely independent of the data used in these two earlier studies. Values are slightly closer to Devred et al. [14] at 443, 490, and 670 nm, and Brewin et al. [31] and Uitz et al. [33] at other wavelengths. High picophytoplankton $a^{*}\left(\lambda \right)$ values at blue wavelengths have been attributed to the presence of non-photosynthetic cartenoids, such as zeaxanthin or β-carotene, that absorb in this region of the spectrum [66]. Picophytoplankton $a^{*}\left(443\right)$ values obtained here (0.183 m${}^{2}$ [mg *C*]${}^{-1}$) are similar to those from monospecific laboratory cultures of *Prochlorococcus* [67,68,69].

Retrieved $a^{*}\left(\lambda \right)$ values for nanophytoplankton (Figure 3d) also agree with those of Brewin et al. [31] and Devred et al. [14], but are slightly lower than Uitz et al. [33] in the blue and red region of the spectrum. Large error bars in retrieved $a^{*}\left(\lambda \right)$ for nanophytoplankton indicate a high natural variability in $a^{*}\left(\lambda \right)$ for this group. It is not surprising considering that nanophytoplankton span a relatively large size range and are known to have high levels of diversity [70]. The high value at 412 nm obtained in this study, relative to other wavelengths, is somewhat surprising when compared with earlier studies. However, large error bars suggest differences are not significant with other wavelengths in the blue region of the spectrum.

Retrieved $a^{*}\left(\lambda \right)$ values for dinoflagellates and diatoms are plotted against results of other studies on microphytoplankton in Figure 3e,f. It is worth considering that, though both dinoflagellates and diatoms are typically microphytoplankton, the microphytoplankton $a^{*}\left(\lambda \right)$ from other studies are not expected to match exactly the dinoflagellates and diatoms $a^{*}\left(\lambda \right)$ from this study, since microphytoplankton $a^{*}\left(\lambda \right)$ represents a combination of both groups. In general, dinoflagellate $a^{*}\left(\lambda \right)$ is higher than the Brewin et al. [31] and Devred et al. [14] studies and diatom $a^{*}\left(\lambda \right)$ lower. Dinoflagellate $a^{*}\left(\lambda \right)$ is higher than all other studies between 490 and 510 nm, but overlaps with Devred et al. [14] in blue regions (412 and 443 nm), agrees with Uitz et al. [33] in green regions (550 to 560 nm) and overlaps with all studies in the red region (665 to 683 nm). Diatom $a^{*}\left(\lambda \right)$ is in good agreement with estimates from Ciotti et al. [12] in blue and green regions of the spectrum (412 to 620 nm) but in better agreement with Brewin et al. [31], Devred et al. [14] and Uitz et al. [33] in the red (665 to 683 nm). Lowest $a^{*}\left(\lambda \right)$ values for diatoms in blue and green wavelengths can be linked to the strong package effect occurring in this group [11]. Values of $a^{*}\left(443\right)$ for diatoms and dinoflagellates are also comparable with laboratory studies on these groups [71,72].

Differences in $a_{i}^{*}\left(\lambda \right)$ among studies (Figure 3) may be due to a variety of reasons. While on the one hand the datasets used in this paper have been used previously by the bio-optical community for studying phytoplankton absorption and community structure [10,11,14,32,59,73,74], and are among the few datasets available for such large spatial-scale analysis, the complication of data collected by different investigators and in different laboratories will always be vulnerable to variations in lab-procedures and techniques [52]. Increasing efforts are being made by the international community to minimise such differences, by improving international protocols for measuring properties like $a_{ph}$ (e.g., [75]). Differences in $a_{i}^{*}\left(\lambda \right)$ among studies (Figure 3) may also be dependent on the methods used to derive phytoplankton composition ($C_{i}$). The model of Brewin et al. [46] was fitted using a combination of High Performance Liquid Chromatography (HPLC) pigment data and sequential size-fractionated chlorophyll measurements. This is different to other studies, for example, Uitz et al. [33] used HPLC data only, and the use of HPLC in their study for deriving phytoplankton composition was not exactly the same as that used in the Brewin et al. [46] study, which introduced modifications to the method [14,34]. The Ciotti et al. [12] study used a combination of methods allocating spectra based on dominant group, and Devred et al. [14] used $a_{ph}$ and *C* to derive $C_{i}$, making assumptions about the composition at extreme (low) ends. Differences in $a_{i}^{*}\left(\lambda \right)$ among studies (Figure 3) may also be related to regional and seasonal differences in the datasets used, and differences in mathematical and statistical methods for determining $a_{i}^{*}\left(\lambda \right)$ (e.g., fitting procedures, variations in the use of parametric and non-parametric statistics).

### 3.2. Model Validation

The validation data were used to verify the performance of the four-population model at retrieving total $a_{ph}\left(\lambda \right)$. The model was tested using both in situ and satellite total chlorophyll-a (*C*) as input. For the in situ validation, correlation coefficients (*r*) ranged from 0.87 to 0.92 and root mean square errors (Ψ) between 0.21 and 0.29 (Table 3), in good agreement with statistical tests performed on the parameterisation data (Figure 2). For the satellite validation, lower *r* values (0.76 to 0.80) and higher Ψ (0.27 to 0.39) were obtained (Table 3). However, results from these statistical tests are influenced by discrepancies between in situ and satellite chlorophyll-a (see footnote to Table 3), and are comparable with other validation studies estimating $a_{ph}\left(\lambda \right)$ from remote sensing reflectance ($R_{rs}$) data [76].

The performance of the four-population model at retrieving $a_{ph}\left(\lambda \right)$ was found to be comparable with results of two other models that estimate $a_{ph}\left(\lambda \right)$ as a function of total chlorophyll-a (*C*) [23,31], using both in situ and satellite total chlorophyll-a (*C*) as inputs (Table 3). There were no significant differences in *r* among models according to the *z*-score (p>0.05, see [76] for computation of *z*-score). There were also no significant differences in Ψ, with 95 % confidence intervals in Ψ overlapping (computed from the standard error and the t-distribution of the sample size), with the exception of 412 nm. Here, the four-population model had significantly lower Ψ than both the three-population model and the power-law model in both validation cases (using in situ and satellite *C* as input). Furthermore, at 443 nm, and in both validation cases, the four-population and three-population model had significantly lower Ψ than the power-law model.

With respect to estimating total $a_{ph}\left(\lambda \right)$ only from total chlorophyll-a (*C*), and considering performance similar to the two simpler models at most wavelengths (that have fewer parameters), the four-population model is a less parsimonious model. However, the four-population model has advantages over the two simpler models, including: (1) its parameters have clear interpretations and the approach ensures plausible values of $a^{*}\left(\lambda \right)$ at extreme chlorophyll-a concentrations, unlike an empirical power function; (2) the model can determine $a_{ph}\left(\lambda \right)$ for four phytoplankton groups, rather than three as in the three-population model; and (3) the model incorporates the effect of changes in SST on total and group-specific $a_{ph}\left(\lambda \right)$, which could be important when modelling phytoplankton absorption in a future, warmer ocean under climate change.

### 3.3. Variations in $a_{ph}\left(\lambda \right)$ with Temperature and Community Structure

The four-population model allows us to explore the impact of changes in SST on phytoplankton community structure and consequently $a_{ph}\left(\lambda \right)$. Figure 4a,b shows the effect of SST on total $a_{ph}\left(443\right)$ and total $a^{*}\left(443\right)$ as functions of total chlorophyll-a (*C*). Larger differences are seen at lower chlorophyll concentrations, where the $a^{*}\left(443\right)$ (and consequently $a_{ph}\left(443\right)$) are higher in warmer waters. This is a consequence of a change in the composition of phytoplankton (Figure 4c,d) from a dominance (in terms of *C*) of picophytoplankton in warmer, low-chlorophyll waters (Figure 4c, picophytoplankton have higher $a^{*}\left(443\right)$, see Figure 3), to nanophytoplankton in cooler, low-chlorophyll waters (Figure 4d, nanophytoplankton have lower $a^{*}\left(443\right)$, see Figure 3). The four-population model constrains $a^{*}\left(443\right)$ at very high and very low chlorophyll-a concentrations, since the range of values of $a^{*}\left(\lambda \right)$ is bounded by the values associated with the populations. However, as the populations change with SST, so do the bounds (Figure 4b). Interestingly, the functional relationship between $a^{*}\left(443\right)$ and *C* is in good qualitative agreement with the hyperbolic tangent function of Carder et al. [28] (Figure 4b). Though biases are observed at the higher end of chlorophyll concentrations, temperate parameters of the Carder et al. [28] model (likely representative of cooler waters) are shifted lower than those from subtropical water (likely representative of warmer waters), consistent with the four-population model at lower chlorophyll concentration (<1 mg m${}^{-3}$).

Estimates of total $a^{*}\left(\lambda \right)$ for a range of total chlorophyll-a concentrations (*C*) are plotted in Figure 5 using the four-population model for three contrasting temperature values: 24 ${}^{\circ }$C; 17 ${}^{\circ }$C; and 10 ${}^{\circ }$C. The change in the upper bound of $a^{*}\left(\lambda \right)$ with SST at low chlorophyll is most apparent in the blue part of the spectrum, but still occurs in the green and red (Figure 5a–c). Figure 5d–o show the fractional contribution to specific absorption of each of the four phytoplankton groups, relative to total $a_{ph}\left(\lambda \right)$, for the same simulations. In general, picophytoplankton control total $a^{*}\left(\lambda \right)$ at low chlorophyll concentration, nanophytoplankton at intermediate chlorophyll, and diatoms and dinoflagellates at high chlorophyll. However, there are clear spectral variations, as well as variations with temperature. Although nanophytoplankton dominate the fractional contribution of each group relative to total chlorophyll at low SST and low chlorophyll (Figure 4a,d), picophytoplankton are more efficient in absorbing light and consequently have a greater influence on $a_{ph}\left(\lambda \right)$ in the same temperature and chlorophyll range.

### 3.4. Towards a Mechanistic Understanding of Temperature in the Four-Population Model

SST influence in the four-population model is represented through the parameters that control the relationship between group-specific chlorophyll ($C_{i}$) and total chlorophyll (*C*) (Equation (5)–(8)): the asymptotic maximum value for cells <20 μm and <2 μm ($C_{1,2}^{m}$ and $C_{1}^{m}$), the fraction of total chlorophyll in two size classes (<20 μm and <2 μm) as total chlorophyll tends to zero ($D_{1,2}$ and $D_{1}$), and the fractional contribution of dinoflagellates ($C_{3}$) and diatoms ($C_{4}$) to microphytoplankton chlorophyll ($C_{3,4}$, cells >20 μm). The extent to which temperature has a direct or an indirect influence on phytoplankton community structure is an area of active research [40,42,77,78,79].

In the favour of direct control, temperature is known to influence the physiology of phytoplankton [80]. As highlighted by López-Urrutia and Morán [78], temperature impacts uptake rates of nutrients, for example, through temperature-dependent metabolic processes that influence nutrient ion handling times, or through modifying nutrient diffusion and fluid viscosity [81]. Temperature also can influence resource allocation; for example, the relative nitrogen-to-phosphorus demands in phytoplankton have been linked to temperature [82]. Phytoplankton loss terms (e.g., grazing) have been related to parameters of the model (e.g., the asymptotic maximum value for small cells [83]) and to temperature. For example, López-Urrutia [84] found that temperature can modify the grazing interactions between phytoplankton and their predators. All these factors will directly influence bottom-up and top-down control on these phytoplankton groups.

Nonetheless, the relationship between temperature and model parameters may simply be an indirect effect of covariation between temperature and resource supply [77,79]. Inverse relationships between temperature and nutrients are well known in the region [85], as are positive relationships between temperature and light availability in the mixed-layer [46]. Temperature may also covary with the spectral quality of light in the mixed layer, which has also been shown to influence phytoplankton community structure [2]. The influence of temperature on model parameters has not only been seen in the North Atlantic study of Brewin et al. [46], but also in other studies, using different datasets and methods, and in different regions, from polar to tropical waters [86,87]. Future work is needed to understand mechanistically how temperature influences model parameters. For example, the evidence and length of lags between temperature and changes in phytoplankton community structure could be useful for understanding direct and indirect effects.

### 3.5. Impact of Variations in $a_{ph}\left(\lambda \right)$ on the Blue-to-Green Ratio of Remote-Sensing Reflectance

We integrated the four-population absorption model into a simple model of ocean colour (see Appendix A). This model was used for the sole purpose of illustrating the influence of changes in SST on phytoplankton community structure, and consequently, the blue-to-green ratio of remote-sensing reflectance. Figure 6 shows the blue-to-green maximum band ratio of remote sensing reflectance ($R_{rs}\left(443>490>510\right)/R_{rs}\left(555\right)$) plotted as a function of total chlorophyll-a (*C*) and SST using this model. Overlain on these simulations is the globally-tuned NASA OC4v6 model [88,89].

Variations in the maximum band ratio at higher chlorophyll (>1 mg m${}^{-3}$) are relatively small and agree well with the empirical OC4v6 model. Larger differences emerge at lower chlorophyll (<1 mg m${}^{-3}$). Here, there is a significant increase in blue light relative to green light with decreasing SST (Figure 6a). This change is characterised by a shift in the composition of phytoplankton from small cells (picophytoplankton) in warmer waters to larger cells (nanophytoplankton and diatoms) in cooler waters (Figure 6b–d). Interestingly, the OC4v6 algorithm tracks the warmer water simulations in these low chlorophyll waters, possibly reflecting the distribution of data in the database used to parameterise the OC4v6 algorithm; after all, low chlorophyll waters are generally more prevalent in subtropical (warmer water) regions. In fact, the median SST (OISST) for HPLC chlorophyll-a concentrations <0.4 mg m${}^{-3}$ in the NOMAD database (Version 2.0 ALPHA, parts of which were used to parameterise OC4v6) is 20 ${}^{\circ }$C, a temperature range where model simulations and OC4v6 agree reasonably (Figure 6a).

Results for the simulations in Figure 6 are consistent with results of some regional studies in cold waters. For example, empirical algorithms that estimate total chlorophyll-a from reflectance ratios typically report higher ratios for the same chlorophyll in the Southern Ocean [90,91], which has been attributed to changes in pigment packaging and a shift toward larger celled phytoplankton [86,92]. Other studies have shown that, in high latitude regions, often characterised by the presence of diatoms, blue-to-green reflectance ratios are higher than in low latitude regions, for similar chlorophyll and coloured dissolved organic matter (CDOM) concentrations [10]. With predicted changes in ocean temperature in the North Atlantic [93], and consequently phytoplankton population, the four-population model could be useful for exploring consequential changes in ocean colour, or verifying bio-optical ecosystem model projections of ocean colour [94].

In fact, it has been suggested [94] that parts of the future ocean could be bluer as phytoplankton concentration and community composition is modified due to climate change. The results presented here (Figure 5 and Figure 6) raise the possibility that temperature-dependent alterations in the phytoplankton absorption properties could counteract any such tendencies to some extent: We see in Figure 6 that the blue-to-green ratio of remote-sensing reflectance decreases with increasing temperature, for a given chlorophyll concentration, and especially for low-chlorophyll waters. To resolve the question of whether such patterns would persist in a future ocean decades away from now, we need to understand better the reasons for a temperature dependence in absorption (see Section 3.4). Another possibility is that higher temperatures are associated with high-light environments, and that the temperature effect observed here is actually a consequence of photo-acclimation: in high-light environment, phytoplankton are expected to have lower chlorophyll per cell than in low-light conditions, which in turn would lead to reduced flattening effect on phytoplankton absorption and higher phytoplankton absorption in the blue relative to green. But such discussions must remain speculative at present, until further investigations can be carried out to test the cause of the observed temperature dependence of phytoplankton absorption.

### 3.6. Mapping Uncertainty in Group-Specific $a_{ph}\left(\lambda \right)$

Figure 7 illustrates an application of the four-population model and Equation (14) in the North East Atlantic, to estimate, from satellite data, $a_{ph}\left(443\right)$ for each group (diatoms, dinoflagellates, nanophytoplankton, and picophytoplankton, Figure 7e,h,k,n) and the uncertainty in these estimates (Figure 7f,i,l,o), from knowledge of the chlorophyll concentrations for each group (Figure 7d,g,j,m), derived from total chlorophyll and SST (Figure 7a,c), uncertainties in the chlorophyll concentrations for each group (from OWT data, Figure 7b, and Table 5 of [46]), and uncertainties in $a^{*}\left(443\right)$ (Table 2). Uncertainty is lowest for picophytoplankton (median 96 %), followed by dinoflagellates (median 121 %), nanophytoplankton (median 124 %), and diatoms (median 140 %). In general, for all groups, there is an increase in uncertainty with increasing total chlorophyll and OWT. This is most pronounced for dinoflagellates. Reducing uncertainty in these estimates requires reducing the uncertainties in the input to Equation (14) (group specific $a^{*}\left(443\right)$ and *C*), which will require improvements in the in situ measurements of phytoplankton group chlorophyll and $a_{ph}\left(\lambda \right)$, and better characterisation of uncertainties associated with differences in the observational scales between satellite and in situ data [95], since uncertainties in group-specific chlorophyll are based on satellite and in situ data match-ups [46].

Increasing efforts to incorporate bio-optical modules into multi-phytoplankton ecosystem models are underway (e.g., [94,96]). These modules open a path to assimilating satellite-based phytoplankton group-specific optical properties, such as phytoplankton absorption. It has recently been demonstrated that significant improvements in the forecasting and reanalysis of biogeochemical indicators by multi-phytoplankton ecosystem models can be achieved by assimilating satellite estimates of group-specific chlorophyll concentration, compared with assimilating only total chlorophyll [47,48]. Furthermore, there are advantages to assimilating optical data directly, rather than concentrations [49,97,98]. The CMEMS project “Optical data modelling and assimilation” seeks to move one step further and assimilate phytoplankton group-specific $a_{ph}\left(\lambda \right)$ into the ERSEM model that is run operationally by the CMEMS MFC of the North West European Shelf-Seas. As demonstrated in Figure 7, such satellite products are becoming available, together with per-pixel uncertainties, which are highly desirable for use in data assimilation, as well as other applications.

## 4. Summary

We use the regional phytoplankton group chlorophyll-a model of Brewin et al. [46] to extend a three-population phytoplankton absorption model [31] to a four-population model. The new model estimates the spectral phytoplankton absorption coefficient ($a_{ph}\left(\lambda \right)$) of four phytoplankton groups (picophytoplankton, nanophytoplankton, dinoflagellates, and diatoms) as a function of the total chlorophyll-a concentration (*C*) and SST. A dataset of $a_{ph}\left(\lambda \right)$ (at 12 wavelengths), *C* and SST measurements, compiled from the surface layer of the North Atlantic, was partitioned into training and validation data, the validation data being matched to satellite ocean-colour observations. The model was fitted to the training data to yield the chlorophyll-specific absorption coefficient ($a^{*}\left(\lambda \right)$) for each of the four groups, which compared well with previous field and laboratory studies.

The model was tested using independent validation data and was found to retrieve total $a_{ph}\left(\lambda \right)$ with a similar performance to two earlier models [23,31], using either in situ or satellite total chlorophyll-a data as input. Unlike these earlier models, the new model can determine $a_{ph}\left(\lambda \right)$ for four phytoplankton groups and includes the influence of changes in SST on total and group-specific $a_{ph}\left(\lambda \right)$. We incorporated the new four-population absorption model into a model of ocean colour to demonstrate the influence of change in SST on phytoplankton community structure, and consequently on ocean colour. We also provide a technique for propagating uncertainty through the model and illustrate it using a satellite image. We expect the model to be useful for optical ecosystem model validation and assimilation exercises and for exploring the influence of temperature change on community structure and consequently ocean colour.

## Figures and Tables

**Figure 1 sensors-19-04182-f001:**
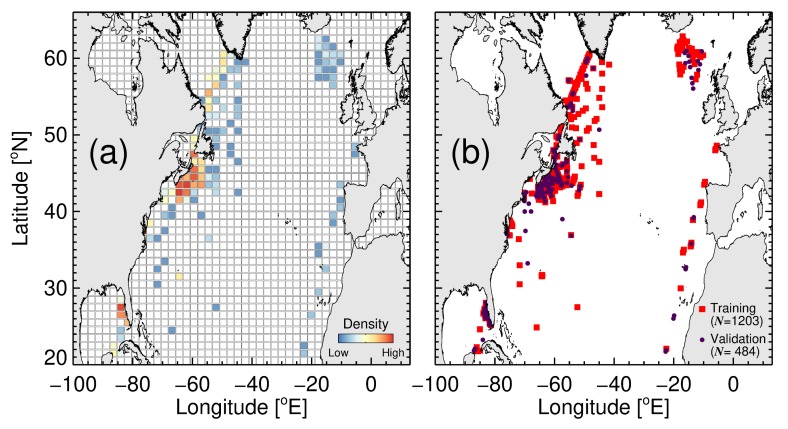
Study site and geographic distribution of the data used in the study. (**a**) Shows the spatial distribution of data used in the study, and (**b**) shows the partitioning into training and validation data.

**Figure 2 sensors-19-04182-f002:**
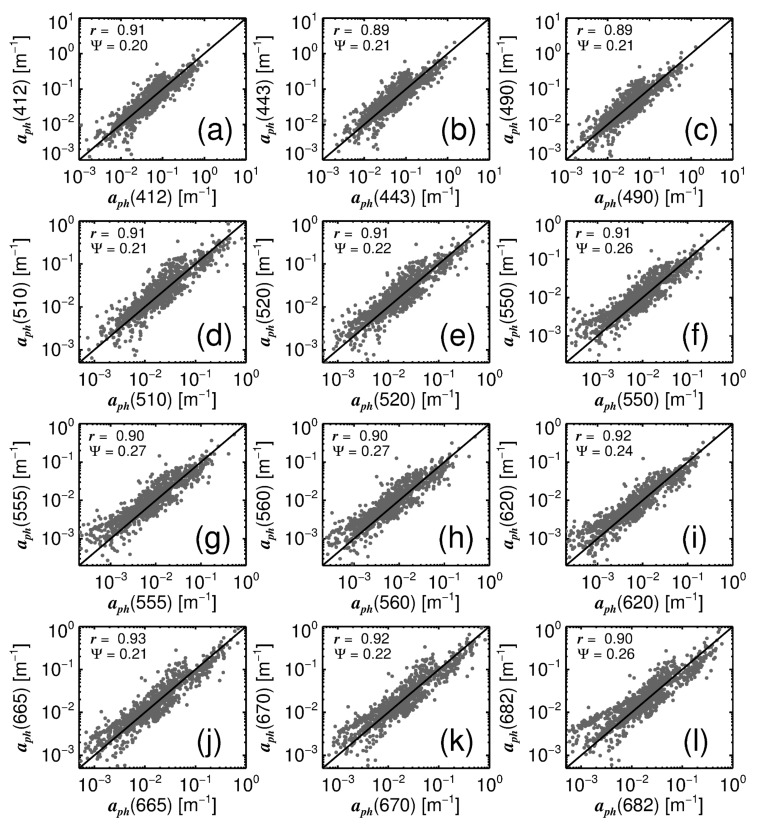
Comparison of modelled $a_{ph}\left(\lambda \right)$ using total chlorophyll (*C*) and sea surface temperature (SST) as input and measured $a_{ph}\left(\lambda \right)$ from the training data. (**a**–**l**) show $a_{ph}\left(\lambda \right)$ scatter plots at each of the 12 wavelengths in the data. Modelled data is on the ordinate and measurements on the abscissa.

**Figure 3 sensors-19-04182-f003:**
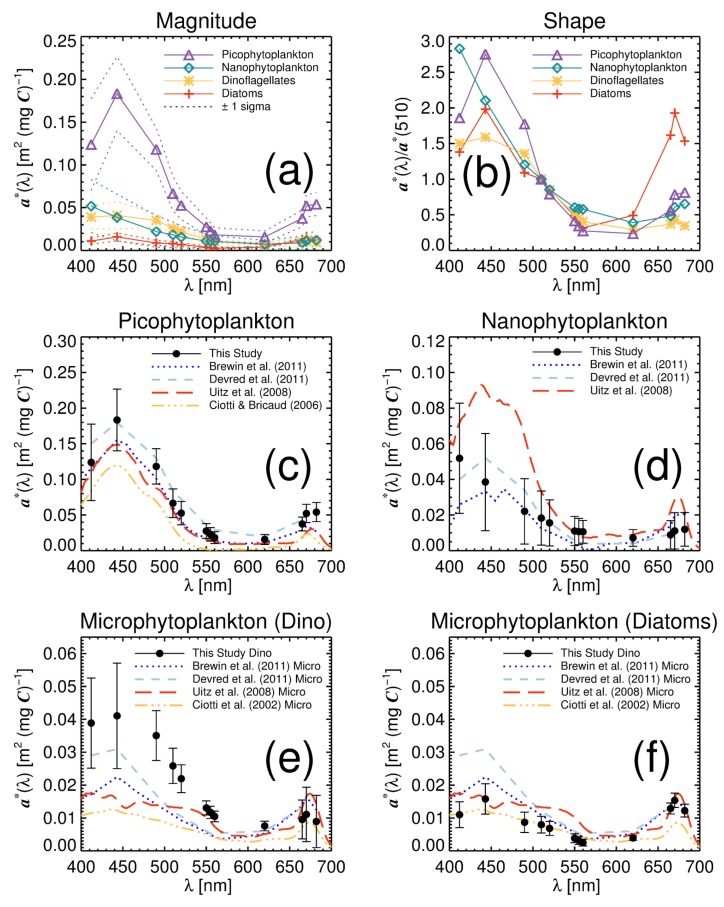
Phytoplankton group chlorophyll-specific absorption coefficients ($a_{i}^{*}\left(\lambda \right)$) derived from tuning the four-population model. (**a**) Magnitude of $a_{i}^{*}\left(\lambda \right)$ for each group and (**b**) spectral form (shape) computed by normalisation of $a_{i}^{*}\left(\lambda \right)$ at 510 nm. Comparison of retrieved $a^{*}\left(\lambda \right)$ values of: (**c**) picophytoplankton with other studies [14,31,33,65]; (**d**) nanophytoplankton with other studies [14,31,33]; (**e**) dinoflagellates with other studies of microphytoplankton [12,14,31,33]; and (**f**) diatoms with other studies of microphytoplankton [12,14,31,33]. Note the different scales of the ordinate axis.

**Figure 4 sensors-19-04182-f004:**
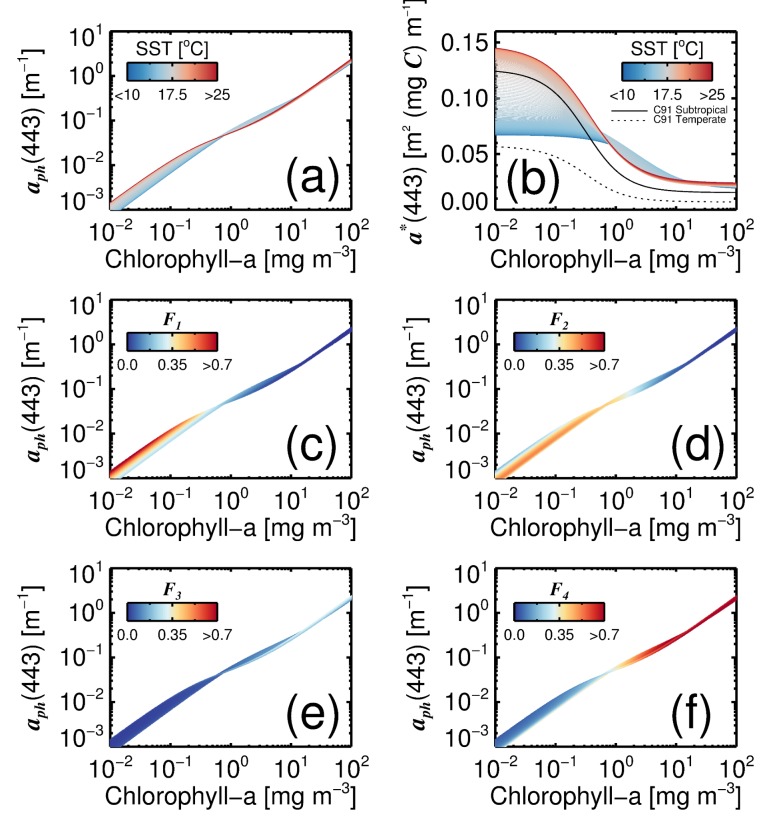
Estimates of $a_{ph}\left(443\right)$ as a function of total chlorophyll-a (*C*) using the four-population model. (**a**) Influence of SST on estimates of $a_{ph}\left(443\right)$ as a function of *C*. (**b**) Influence of SST on estimates of $a^{*}\left(443\right)$ as a function of *C*: C91 refers to the model of Carder et al. [28], for subtropical and temperate waters. (**c**–**f**) The fractions ($F_{i}$) of each phytoplankton group (1 = picophytoplankton, 2 = nanophytoplankton, 3 = dinoflagellates, and 4 = diatoms) relative to *C* for the same model simulations.

**Figure 5 sensors-19-04182-f005:**
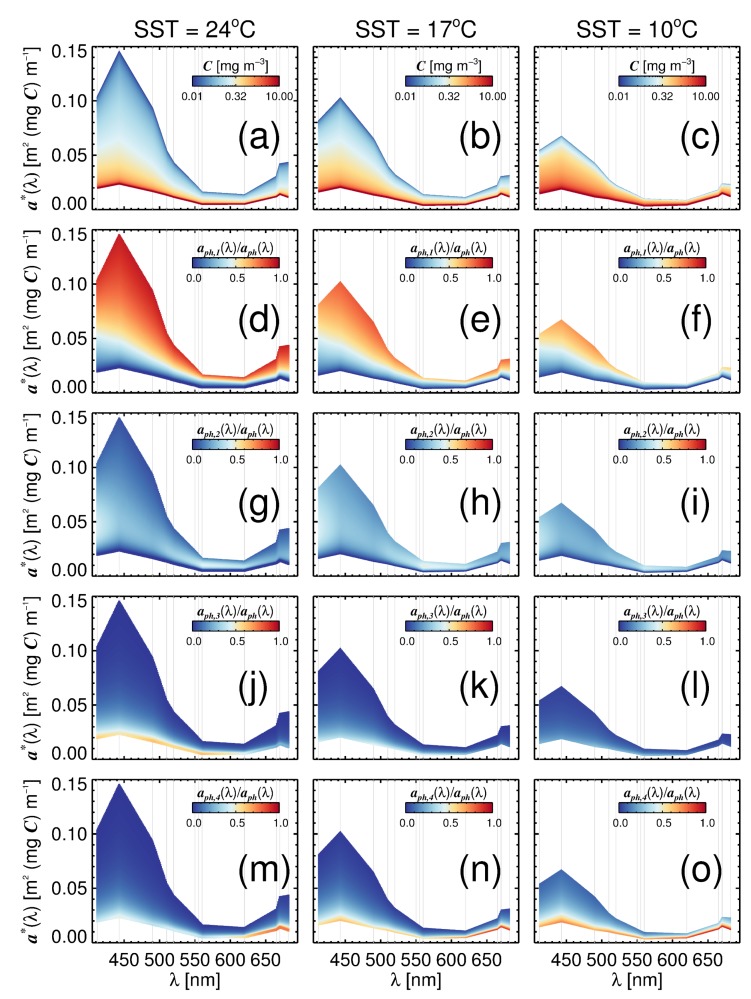
Estimates of $a^{*}\left(\lambda \right)$ as a function of total chlorophyll-a (*C*) using the four-population model at three contrasting temperature ranges: (**a**,**d**,**g**,**j**,**m**) at 24 ${}^{\circ }$C; (**b**,**e**,**h**,**k**,**n**) at 17 ${}^{\circ }$C; and (**c**,**f**,**i**,**l**,**o**) at 10 ${}^{\circ }$C. (**a**–**c**) $a^{*}\left(\lambda \right)$ for a given total chlorophyll-a (*C*) at the three temperature ranges. (**d**–**o**) The fractional contribution of each group (1 = picophytoplankton, 2 = nanophytoplankton, 3 = dinoflagellates and 4 = diatoms) relative to $a_{ph}\left(\lambda \right)$ for each simulation at the three temperature ranges: (**d**–**f**) picophytoplankton; (**g**–**i**) nanophytoplankton; (**j**–**l**) dinoflagellates; and (**m**–**o**) diatoms. Thin grey lines represent wavelengths in the model, all other wavelengths are estimated from linear interpolation between neighbouring wavebands and should be interpreted cautiously.

**Figure 6 sensors-19-04182-f006:**
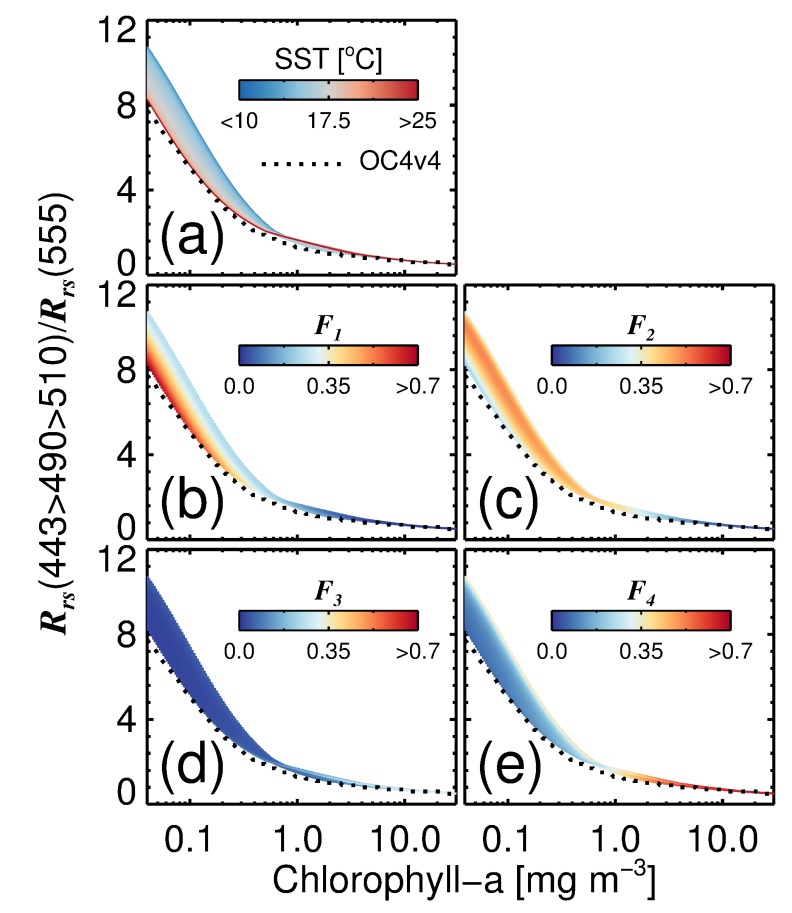
The blue-to-green maximum band ratio of remote sensing reflectance ($R_{rs}$) plotted as a function of total chlorophyll-a (*C*) and SST using a model of ocean colour that integrates the four-population absorption model (see Appendix A). (**a**) Impact of variations in SST on estimates of the maximum band ratio. (**b**–**e**) The fractions ($F_{i}$) of each phytoplankton group (1 = picophytoplankton, 2 = nanophytoplankton, 3 = dinoflagellates, and 4 = diatoms) relative to *C* for the same model simulations. Dashed line represents the NASA OC4v6 model [88,89].

**Figure 7 sensors-19-04182-f007:**
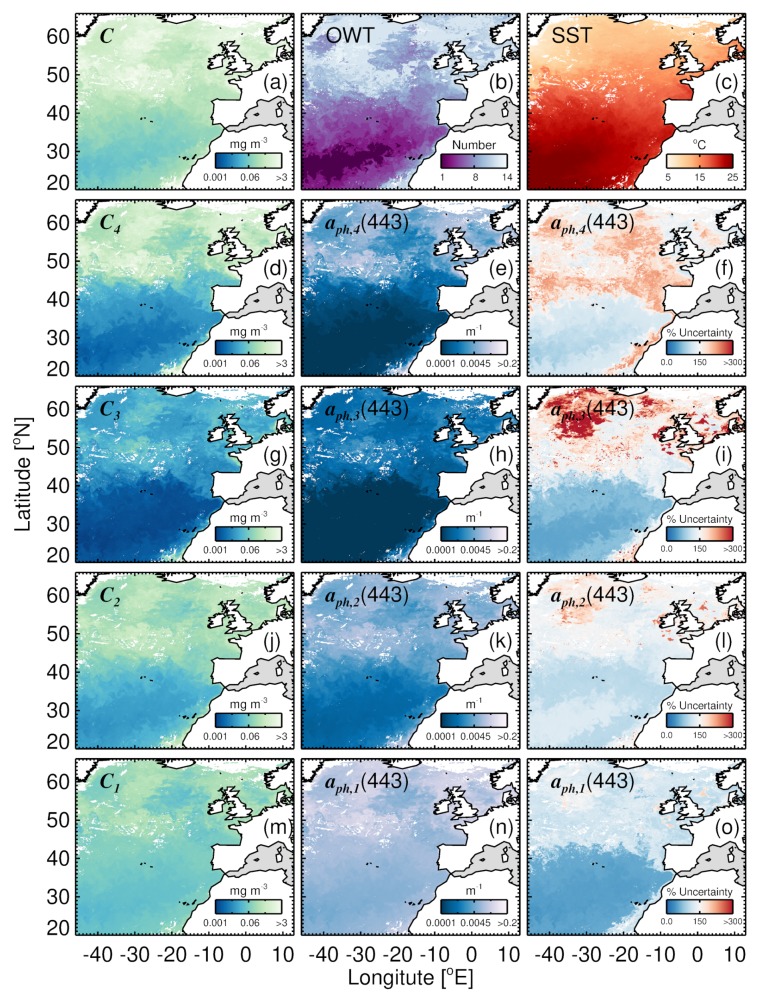
Satellite estimates of phytoplankton group chlorophyll, $a_{ph}\left(443\right)$ and per-pixel errors in $a_{ph}\left(443\right)$ for an eight day composite (17 to 24 June 2008) of Ocean Colour Climate Change Initiative (OC-CCI) chlorophyll and NOAA Optimal Interpolation Sea Surface Temperature (OISST) SST using the four-population absorption model and Equation (14). (**a**) Total chlorophyll, (**b**) dominant optical water type (OWT), and (**c**) SST data. These are used as input to the four-population absorption model to predict: (**d**) diatom chlorophyll ($C_{4}$); (**e**) diatom absorption at 443 nm ($a_{ph,4}\left(443\right)$); (**f**) % uncertainty in $a_{ph,4}\left(443\right)$; (**g**) dinoflagellate chlorophyll ($C_{3}$); (**h**) dinoflagellate absorption at 443 nm ($a_{ph,3}\left(443\right)$); (**i**) % uncertainty in $a_{ph,3}\left(443\right)$; (**j**) nanophytoplankton chlorophyll ($C_{2}$); (**k**) nanophytoplankton absorption at 443 nm ($a_{ph,2}\left(443\right)$); (**l**) % uncertainty in $a_{ph,2}\left(443\right)$; (**m**) picophytoplankton chlorophyll ($C_{1}$); (**n**) picophytoplankton absorption at 443 nm ($a_{ph,1}\left(443\right)$); and (**o**) % uncertainty in $a_{ph,1}\left(443\right).$

**Table 1 sensors-19-04182-t001:** Parameter values for Equation (5)–(8). Taken from Table 4 of Brewin et al. [46].

Model Parameter	Parameters Values ^$^			
	i=a	i=b	i=c	i=d
$G_{i}$ (Equation (5))	−1.51	−1.25	14.95	0.25
	(−1.57↔ −1.43)	(−1.41↔ −1.25)	(14.87↔15.05)	(0.23↔0.26)
$H_{i}$ (Equation (6))	0.29	3.05	16.24	0.56
	(0.28↔0.30)	(2.87↔3.26)	(16.19↔16.29)	(0.55↔0.57)
$J_{i}$ (Equation (7))	0.370	1.13	14.89	0.569
	(0.367↔0.373)	(1.10↔1.16)	(14.87↔14.91)	(0.566↔0.571)
$K_{i}$ (Equation (8))	0.503	1.33	17.31	0.258
	(0.501↔0.505)	(1.31↔1.37)	17.28↔17.32)	(0.256↔0.259)

${}^{\$}$ Bracket values refer to the 2.5% and 97.5% confidence intervals.

**Table 2 sensors-19-04182-t002:** Chlorophyll-specific absorption coefficients (m${}^{2}$ [mg *C*]${}^{-1}$) retrieved from fitting the four-population model (Equation (10)) to the parameterisation data.

Wavelength	Picophytoplankton	Nanophytoplankton	Dinoflagellates	Diatoms
λ (nm)	$a_{1}^{*}$	$a_{2}^{*}$	$a_{3}^{*}$	$a_{4}^{*}$
412	0.124 (±0.054)	0.052 (±0.031)	0.039 (±0.014)	0.011 (±0.004)
443	0.183 (±0.043)	0.039 (±0.027)	0.041 (±0.016)	0.016 (±0.005)
490	0.118 (±0.025)	0.022 (±0.018)	0.035 (±0.008)	0.009 (±0.003)
510	0.067 (±0.020)	0.018 (±0.015)	0.026 (±0.005)	0.008 (±0.003)
520	0.053 (±0.016)	0.016 (±0.013)	0.022 (±0.004)	0.007 (±0.002)
550	0.028 (±0.010)	0.011 (±0.008)	0.013 (±0.002)	0.004 (±0.001)
555	0.023 (±0.009)	0.011 (±0.007)	0.012 (±0.002)	0.003 (±0.001)
560	0.018 (±0.008)	0.011 (±0.006)	0.010 (±0.002)	0.003 (±0.001)
620	0.016 (±0.007)	0.007 (±0.005)	0.008 (±0.001)	0.004 (±0.001)
665	0.037 (±0.010)	0.009 (±0.008)	0.010 (±0.006)	0.013 (±0.002)
670	0.052 (±0.013)	0.011 (±0.010)	0.011 (±0.008)	0.015 (±0.002)
682	0.054 ±0.013)	0.012 (±0.009)	0.009 (±0.008)	0.012 (±0.002)

Bracketed values refer to robust standard deviations.

**Table 3 sensors-19-04182-t003:** Comparison of the performance of the four-population model at retrieving $a_{ph}\left(\lambda \right)$ with the model of Brewin et al. [31] and Bricaud et al. [23] using the validation data, and both in situ and satellite chlorophyll-a data as input.

Wavelength (nm)	In Situ Chlorophyll-a as Input *						Satellite Chlorophyll-a as Input *					
	This Study		Brewin et al. [31]		Bricaud et al. [23]		This Study		Brewin et al. [31]		Bricaud et al. [23]	
	r	Ψ	r	Ψ	r	Ψ	r	Ψ	r	Ψ	r	Ψ
412	0.89	0.21	0.88	0.23	0.89	0.26	0.80	0.28	0.80	0.31	0.80	0.34
443	0.87	0.22	0.86	0.22	0.87	0.25	0.78	0.27	0.78	0.29	0.78	0.32
490	0.87	0.21	0.86	0.21	0.86	0.22	0.77	0.27	0.78	0.27	0.78	0.29
510	0.89	0.21	0.89	0.21	0.89	0.22	0.79	0.28	0.80	0.29	0.80	0.30
520	0.90	0.21	0.90	0.21	0.91	0.22	0.80	0.29	0.81	0.30	0.81	0.31
550	0.90	0.26	0.90	0.25	0.91	0.24	0.80	0.34	0.80	0.36	0.81	0.33
555	0.90	0.26	0.89	0.26	0.91	0.24	0.80	0.35	0.80	0.37	0.81	0.34
560	0.90	0.27	0.89	0.27	0.91	0.24	0.80	0.35	0.80	0.38	0.82	0.33
620	0.91	0.25	0.91	0.23	0.91	0.23	0.79	0.36	0.80	0.35	0.81	0.34
665	0.92	0.22	0.92	0.22	0.92	0.21	0.80	0.34	0.81	0.33	0.81	0.34
670	0.91	0.23	0.92	0.22	0.92	0.22	0.79	0.34	0.80	0.33	0.80	0.34
682	0.88	0.29	0.89	0.26	0.89	0.26	0.76	0.39	0.78	0.37	0.78	0.37

* All statistical tests are performed on log${}_{10}$-transformed $a_{ph}\left(\lambda \right)$ data. A statistical comparison between log${}_{10}$-transformed in situ and satellite chlorophyll-a yielded r=0.80 and Ψ=0.34.

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
