# Peer review of "The Influence of Temperature and Community Structure on Light Absorption by Phytoplankton in the North Atlantic"

_sensors, 2019, doi:10.3390/s19194182_

Round 1

Reviewer 1 Report

This manuscript develops a new model to retrieve absorption coefficients of picophytoplankton, nanophytoplankton, dinoflagellates and diatoms. The model uncertainty is also analyzed. Basically, I found the studies in the manuscript are interesting with the novel knowledge to the marine remote sensing community. I suggest to accept the manuscript after minor revisions. It is better to add a figure showing relationship between Satellite Chlorophyll-a and In-situ Chlorophyll-a, which may display how consistency between the satellite observation and the in-situ data is. 

Author Response

We thank reviewer 1 for taking the time to read and review our paper. We are glad that they found our study to be interesting, novel and acceptable for publication subject to minor revisions.

We also thank the reviewer for raising the point regarding consistency between satellite Chlorophyll-a and in-situ Chlorophyll-a. In light of this comment, and considering the high number of figures already included in the manuscript, we have modified the footnote to Table 2 of the submitted paper (now Table 3 in the revised paper) and added validation statistics for a comparison between satellite Chlorophyll-a and in-situ Chlorophyll-a. This way a reader can see the performance of the satellite Chlorophyll-a data through reference to the Table. We have also added a line to the manuscript highlighting the consistency between satellite Chlorophyll-a and in-situ Chlorophyll-a and stating it will impact statistical tests in the satellite validation of phytoplankton absorption (see lines 272-273 of the revised manuscript).

Reviewer 2 Report

This manuscript fits in the line of articles by Brewin et al. that take size class of phytoplankton as a major discriminator of optical properties, like scatter, backscatter and absorption. Indeed, there is an urgent need, both from model- and from remote sensing- perspective to measure the phytoplankton composition in the world oceans. After reading this manuscript in detail, including the article of Brewin et al. in 2017 (Frontiers in Marine Science), I find the results presented on the inclusion of 4 groups and SST not convincing to warrant publication. Also, the fundamental approach taken in this manuscript should be presented to a journal dealing with microbiology; It is crucial that this work is fully understood at the phytoplankton biophysical level, before the results are translated to a journal like Sensors.

Let me first express that the introduction is written rather well, has a good list of references and is clear in how the data are collected and separated in development- and validation-set. The algebra to separate the groups needs the previous articles to be understood. The manuscript has some major problems, however:

Major comments:

SST has been added to the 3/4-population phytoplankton model in order to constrain and better explain the variation in group chlorophyll concentration and group specific absorption. However, it is not clear from the introduction why SST could be a governing factor of these parameters. I miss a physiological explanation for this relationship. I could also not find an explanation in Brewin et al 2017 – fmars

The relationship between group chlorophyll concentration/group specific absorption and SST might be confounded by nutrient concentrations. It would paint a clearer picture if nutrient data (if available for sampling stations) was aligned with SST to show relationship and covariation.

The separation of micro-phytoplankton group into dino’s and diatoms does not add to more explanatory power of the model. The authors argue that although the four-population model is a less parsimonious model than the three-population model, it allows to determine an extra functional group. I feel like that increasing model complexity is only a valid argument when it does not increase uncertainty in the model parameters compared to the three-population model. I miss a quantification of model parameter uncertainties/ overfitting metrics. The authors do realize that SST has no detectable impact on the model performance, but fail to draw the conclusion that this takes the carpet right under their feet.

Although the authors provide a positive explanation of Fig. 3, I notice major difference with the absorption properties presented in other papers. This makes me wonder if the collection of data is not hampered by systematic differences in lab-procedures, especially at extremely low and extreme high concentrations. The error bars of aph (lambda) are very large and the median curves show some surprising results: for example the curves of Nano- and Dino’s are almost identical (given the large error bars). For example the range 550-600 nm shows a difference of a factor 4 between the groups, while Bricaud always finds that this part of the spectrum shows no major dependence on concentration. In fact the Bricaud relation performs better in the validation test.

In case this manuscript is submitted again: Minor comments:

It is important to explain the reason for the separation of micro into dino’s and diatoms; ERSEM requirements, see in Brewin et al 2017 – fmars

The paper is hard to read due to the overload of parameter abbreviation in the text. It makes it hard to follow the story. I would suggest trying to rewrite the text in a clear way or adding a table for quick reference to parameter definitions…

What is the SST distribution in your dataset? Is it biased to low/high SST? etc etc… This would make clear the SST bias in your model…

Figure 1: many points are overlaid. Consider using a density plot to show low/high concentrations of data points on the map – this clarifies for the reader the bias in sampling data

caption figure 2: ‘… SST as input and measured aph from training data’ – shouldn’t this be validation data?

If there is an effect of SST on phytoplankton absorption – would you not expect a lag between changes in SST and aph? Therefore wouldn’t it be better to see if averaged SST from x previous days are better correlated with aph?

Figure 4-6: color bar for SST, C and fraction F are too similar and make it confusing. I would suggest using different color schemes for these 3 variables to prevent confusing with the reader.

Figure 7: color bar confusing again. White color to indicate high C is confusing. Same for absorption. Also color bar of uncertainty does not make sense. For all these subplots: in general I would associate a white color with absence. But this might be personal.

Author Response

Please find our response to comments raised by Reviewer 2 in the attached PDF file.

Reviewer 3 Report

The manuscript „The influence of temperature and community structure on light absorption by phytoplankton in the North Atlantic concerns very important aspect on the influence of changes in SST on phytoplankton community structure incorporating a new model with a very good previous basis, and consequently, the blue-to-green ratio of remote-sensing reflectance. Such results are worth to publish in Sensors. However, a minor revision is needed, primarily of abstract and English grammar. Please think if the citing the references in abstract is really necessary.

Author Response

We thank reviewer 3 for taking the time to review of our paper. We are glad they believe our study is worthy of publication in Sensors.  

Regarding citing of references in the abstract, we have checked on the “instructions for authors” webpage for the journal (https://www.mdpi.com/journal/sensors/instructions) and found there to be no statement saying references are not allowed in the abstract. Whereas it is not common to include references in the abstract, we felt it necessary in our paper for the following reasons.

1) In the first instances of references (line 4 and 5), there are other three-component models in the literature (e.g. Devred et al. 2011. A three-component classification of phytoplankton absorption spectra: Application to ocean-color data. Remote Sensing of Environment115(9), pp.2255-2266.; Sathyendranath et al 1989. A three-component model of ocean colour and its application to remote sensing of phytoplankton pigments in coastal waters. International Journal of Remote Sensing10(8), pp.1373-1394) so for clarity we cited the three-component model we were extending, and the model we were using to do this (Brewin et al. [1] and [2]).

2) In the second instance of references in the abstract (line 13), there are a wide variety of other models that relate total chlorophyll to phytoplankton absorption in the literature, so to be clear we cited the models we used for the comparison (Bricaud et al. [3] and Brewin et al. [2]).

Of course, if the editor and journal feel it to be inappropriate to use references in the abstract, we would be willing to modify our abstract and remove the references, but would rather keep them for the sake of clarity if possible.

We have carefully re-read our paper and corrected grammatical errors we spotted.